# Herpes Simplex Virus and Varicella Zoster Virus Infections in Cancer Patients

**DOI:** 10.3390/v15020439

**Published:** 2023-02-05

**Authors:** Ralph Tayyar, Dora Ho

**Affiliations:** Division of Infectious Diseases and Geographic Medicine, Department of Medicine, Stanford University School of Medicine, Stanford, CA 94305, USA

**Keywords:** herpes simplex virus, varicella zoster virus, herpes zoster, cancer, immunocompromised

## Abstract

Herpes simplex virus (HSV) and varicella zoster virus (VZV) are alpha herpesviruses that establish life-long latent infection in neuronal ganglia after primary infection. Periodic reactivation of these viruses results in recurrent infections that can have significant impact on patients’ quality of life. HSV commonly causes oral and genital mucocutaneous infections whereas VZV is responsible for varicella/chickenpox and herpes zoster/shingles, but cancer patients are at particularly higher risk of complications including disseminated and visceral infections due to impaired cell-mediated immunity. While diagnosis of more common HSV and/or VZV infections is frequently clinically based, immunocompromised hosts may have atypical skin presentation or visceral involvement. Thus, diagnostic confirmation using virus-specific tests such as polymerase chain reaction or immunohistochemical staining is crucial in some cases. Oral acyclovir, valacyclovir and famciclovir are usually used for mild to moderate infections and intravenous acyclovir is the drug of choice for severe or disseminated infections. Foscarnet can be used when acyclovir-resistance is confirmed or suspected. Pharmaceutical prophylaxis against HSV and/or VZV should be considered in high-risk cancers patients. Currently, there is no commercially available vaccine against HSV, but VZV vaccines are available to prevent varicella and zoster.

## 1. Introduction

Both Herpes Simplex Virus (HSV) and Varicella Zoster Virus (VZV) belong to the Herpesviridae family, whose members are large, enveloped viruses with a linear double-stranded DNA genome in an icosahedron capsid. Eight herpesviruses are known to cause infection in humans, including HSV-1, HSV-2, VZV, cytomegalovirus (CMV), human herpesvirus (HHV)-6, HHV-7, Epstein–Barr virus (EBV), and Kaposi sarcoma-associated herpesvirus (KSHV, also called HHV-8). Based on the biologic properties of these viruses, they are sub-divided into three sub-families, the Alphaherpesvirinae (HSV-1, HSV-2, VZV), Betaherpesvirinae (CMV, HHV-6, HHV-7), and Gammaherpesvirinae (EBV, KSHV) [1]. 

The alpha herpesviruses are notable for their predilection to establish latent, life-long infection in dorsal root sensory ganglia, then reactivate periodically to cause asymptomatic viral shedding or clinical disease. Such characteristics of alpha herpesviruses undoubtedly shape the clinical manifestations that we observe with HSV and VZV [1]. 

These viruses can cause infection of mucocutaneous tissues and visceral organs, resulting in a wide spectrum of clinical diseases, from asymptomatic shedding to fulminant sepsis or death. The course and outcome of the infection largely depend on the interplay between the virus and the host, including anatomic site(s) involved, viral type (HSV-1 vs. HSV-2) as well as age and immune status of the host [2,3]. Cancer patients, particularly those who received myelosuppressive chemotherapy or underwent hematopoietic cell transplantation (HCT), are at increased risk of complications from these viruses [4,5].

Infection of these viruses can have a substantial impact on quality of life. It can cause significant morbidities and even mortalities, particularly among those with immunocompromised conditions, including patients with malignancies. This review will discuss the epidemiology, biology, clinical manifestations, treatment, and prevention of HSV and VZV, with particular focus on cancer patients.

## 2. Epidemiology

### 2.1. HSV-1 and HSV-2

Both HSV-1 and HSV-2 have a huge global burden. Per World Health organization (WHO), an estimated 3.7 billion people, around 66.6% of the world’s population aged 0–49 were infected with HSV-1 in 2016, while an estimated 491.5 million people equivalent to 13.2% of the world’s population aged 15 to 49 years, were infected with HSV-2 [6]. From a 2015-16 survey, the prevalence of HSV-1 and HSV-2 in the United States (US) population of age 14–49 were 47.8% and 11.9%, respectively (Centers for Disease Control and Prevention, CDC) [7].

Seroprevalence of HSV-1 and HSV-2 in general increases with age and varies among different populations (CDC) [7]. HSV-1 is usually acquired early in childhood, while seroconversion of HSV-2 starts around puberty with the initiation of sexual activities. HSV-1 is mainly transmitted by oral-to-oral contact, whereas HSV-2 is almost exclusively transmitted sexually, causing genital herpes [8]. However, HSV-1 can also be transmitted to the genital area through oral-genital contact, accounting for ~140 million cases of genital herpes in one study [9]. Similarly, oral shedding of HSV-2 has also been reported [10,11].

Various studies have shown a higher prevalence of HSV infection or reactivation among cancer patients [9,12,13,14,15]. For instance, the reactivation of HSV-1 was significantly associated with chemo-induced oral mucositis; however, whether HSV plays a major causative role in oral mucositis remains inconclusive [12]. Similarly, although HSV has been isolated from tissue specimens of oral squamous cell carcinoma, esophageal cancer and cervical cancer [13,14,15], its role in carcinogenesis has not be proven. Of interest, in patients with different types of cancer undergoing central nervous system irradiation, prevalence of HSV encephalitis was shown to be higher than the general population [16].

### 2.2. VZV

Varicella or chickenpox represents the primary infection by VZV, after exposure of susceptible hosts to the virus. Chickenpox is extremely contagious and is thought to be transmitted through the respiratory route. Similar to HSV, VZV also establishes life-long latency in neuronal tissues, and its reactivation leads to zoster or shingles. Chickenpox affects children worldwide, but for countries that adopt vaccination programs against chickenpox, the epidemiology of VZV has changed dramatically [3]. Historically, the incidence of varicella mirrored the birth rate. After the implementation of universal childhood vaccination, the incidence of varicella declined by 90% and related mortality declined by 66% [17]. In 2006, the Advisory Committee on Immunization Practices (ACIP) started to recommend two doses of varicella vaccine for children as opposed to one dose, as well as all susceptible adolescents and adults [18]. Following the implementation of two doses, the incidence of varicella further decreased from 48.7 cases/100,000 person in 2005 to 24.5 in 2008 [19]. 

For herpes zoster (HZ), it is estimated that about one out of every three people in the US will develop this viral infection, with roughly 1 million people affected per year. HZ can occur at all ages, although it mainly affects elderly adults with incidence increasing with advancing age [20]. Much of the increase begins at age 50–60 years, up to 11 per 1000 person years among those older than 80 years old [21]. The lifetime incidence rate of HZ is estimated at 10–20% in the general population and as high as 50% of those surviving to age 85 years [22]. The risk of HZ is also increased among immunocompromised hosts. For purposes of comparison, the estimated annual incidence of HZ in adults without underlying disease is 0.5%, as compared to about 7% among solid organ transplant recipients and 16–63% among those with HCT, up to 80% among cord blood transplant recipients [5]. In a large prospective cohort study of >240,000 adults in Australia from 2006 to 2016, participants with hematologic or solid cancer had significantly higher HZ than those without cancer (adjusted hazard ratio 3.74 and 1.30, respectively) [23]. More than 20 cases of HZ per 1000 person years have been reported in patients with any hematologic malignancies, with the highest rates amongst those with monocytic leukemias, multiple myeloma, and Hodgkin lymphoma. Among patients with solid tumors, those with gastric, brain, and ovarian cancers had the highest HZ incidence [24].

Most individuals experience only one episode of HZ; only a few percentages of patients will have more than one episode. In fact, recurrent cutaneous lesions are much more likely caused by HSV than by VZV. However, the risk of recurrent HZ is increased among immunocompromised patients [3]. 

## 3. Biology and Replication 

A detailed description of the biology of these alpha herpesviruses is beyond the scope of this article and is reviewed elsewhere [1,8,25]. Briefly, the virions of herpesviruses all share similar morphology and structure: A core containing a single double-stranded DNA genome is enclosed by an icosahedral capsid, which is further surrounded by an amorphous protein coat called the tegument. The outermost layer of the virion is a lipid envelope containing various viral glycoproteins [1].

The HSV genome is about 150 kilo-basepair (kbp) and encodes about 84 proteins [2]. The overall sequence homology between HSV-1 and HSV-2 is about 50% [26]. Upon attachment of the virion glycoproteins to host receptors, fusion of the viral envelope and cell membrane occurs resulting in the release of the nucleocapsid (and the tegument) into the cytoplasm. The nucleocapsid is further transported to the nuclear pores, through which viral DNA is released into the nucleus. Sequential transcription of various viral genes then ensues, followed by the production of viral proteins and replication of the viral genome. The viral DNA is packaged into pre-formed capsids and the newly formed nucleocapsid then bud through the inner nuclear membrane and traverse the cytoplasm through the endoplasmic reticulum and Golgi apparatus [2]. The egress process is thought to involve a pathway of envelopment and de-envelopment. Viral spread likely occurs via two major routes: in cell–cell spread, by which progeny virions are delivered directly to cellular junctions to infect adjacent cells, or in cell-free release, with progeny virions released into the extracellular milieu, potentially allowing the infection of distant cells [27].

VZV genome is about 125 kpb in size, encoding ~75 proteins. The process of viral replication is thought to be similar to that of HSV. VZV is highly cell associated and direct cell to cell spread is the major mechanism of viral multiplication [3]. However, its tropism for T cells is thought to play a crucial role in the dissemination of the virus, during the viremic phase of infection, from the initial sites of infection to the rest of the body [25].

## 4. Immunology and Host Responses

The innate immunity is the first line of defense mechanism against HSV and VZV infections. This includes natural killer cells, macrophages, dendritic cells, and Toll-like receptors. However, the adaptive immunity (i.e., cell-mediated and humoral responses), plays the most important role in controlling these viral infections. Despite adequate levels of neutralizing antibodies against HSV and VZV, reactivation of viruses can still occur, indicating that cell-mediated immunity is crucial in controlling the severity and spread of primary infections and in reducing re-activations. In particular, the level of plasma HSV-specific CD8+ cytotoxic T cell precursors has been shown to be the best predictor for frequency and severity of viral reactivations [2,3]. Hence, cancer patients, especially those receiving T-cell depleting therapies and HCT, are at higher risk of more severe forms of HSV/VZV reactivation and disseminated disease; thus, prophylaxis is essential, as discussed below. It has been demonstrated that pediatric cancer patients may lose pre-existing humoral immunity against vaccine-preventable diseases (including VZV) after chemotherapy or HCT, thus underlining the need for re-vaccination post-chemotherapy [28] or after HCT.

## 5. Pathogenesis and Clinical Manifestations

### 5.1. HSV-1 and HSV-2

Primary infection with HSV starts with exposure of the virus to mucosa or abraded skin. After viral entry and replication in cells of the epidermis and dermis, the released virions can traverse the neuroepithelial junction and gain entry into nerve endings of the sensory neurons. The virus then reaches the neuronal cell bodies within the sensory ganglion by retrograde transport, where it can establish life-long latency. These infected neuronal cells can maintain normal cell function, while harboring the viral genome in a latent or repressed state with transcription of only a few viral RNAs. For HSV-1 infection, the trigeminal ganglia are the most common site of latency; for HSV-2 the sacral nerve root ganglia are most frequently involved. The mechanisms that control and maintain viral latency are unclear, but a virus can reactivate periodically without apparent damage of the infected neurons, travel back to the nerve endings, and infect cells of the mucocutaneous tissues resulting in either asymptomatic shedding, or recurrent infection manifested as clinical disease [2].

Many clinical diseases have been associated with HSV. For both subtypes, primary infection (e.g., gingivostomatitis with HSV-1 and genital herpes with HSV-2) tends to be more severe than subsequent recurrences (as orolabial herpes and recurrent genital herpes, respectively). While both HSV-1 and HSV-2 can cause genital herpes, recurrences tend to be more frequent with HSV-2, likely representing evolutionary adaptation of HSV-2 to the sacral ganglia milieu. Reactivation of latent HSV is exceptionally frequent in cancer patients, although virus reactivation seems to occur independently of cancer chemotherapy [29].

A number of complications have been associated with HSV infection [2], from keratitis and Bell’s palsy to encephalitis (noted that these entities can also be caused by VZV). In the US, HSV-1 keratitis is the most common cause of infectious corneal blindness and HSV-1 encephalitis is the most common cause of acute viral encephalitis. Moreover, genital herpes (HSV-2) complications include aseptic meningitis, transverse myelitis, and sacral radiculopathy. Other clinical syndromes associated with HSV, such as esophagitis, hepatitis, pneumonitis, and disseminated infection are relatively rare, and manifest mostly in immunocompromised hosts. 

Visceral organ infections can involve any sites, including the esophagus, liver, or lungs. When a single organ is involved, local HSV reactivation or extension from adjacent infection (e.g., vagus nerve reactivation or extension from oropharynx as in the case of HSV esophagitis) may play a role. However, viremia may herald visceral infection, particularly with multiple-organ involvement or disseminated disease, which can be associated with high mortality in severely immunosuppressed patients. 

#### 5.1.1. Esophagitis

HSV can infect any part of the alimentary tract with the esophagus being most frequently affected. HSV esophagitis can be seen in immunocompetent individuals, but is much more common in immunosuppressed patients, including those with malignancy and those undergoing steroid or immunosuppressive therapy. The “stress” of cancer diagnosis and of treatment (in the form of surgery, chemotherapy, and/or radiation therapy) frequently leads to HSV reactivation, which causes oral or perianal lesions that may mimic or exacerbate chemo-induced mucositis. For those patients with profound immunosuppression, such as those undergoing HCT, ulceration from HSV can be unremitting and disabling, causing significant impact to the patient’s quality of life.

Typical symptoms include dysphagia and odynophagia; some may also experience gastrointestinal bleeding. Endoscopic examination usually demonstrates erosions or ulcers, typically involving the mid to distal esophagus. Since a number of infectious and non-infectious entities can cause ulceration of the esophagus, confirmation of HSV esophagitis requires immunohistochemical staining of biopsy specimens using specific antibodies against HSV. Histopathologic observation of multinucleated giant cells and/or inclusion bodies in the clinical specimen is inadequate to confirm the HSV diagnosis since other herpesviruses (such as CMV or VZV) can also cause esophagitis and produce similar cytopathic changes. In a recent case series of HSV esophagitis from a single center in Taiwan [30], 47 patients with HSV esophagitis were identified histologically from among 1843 patients with esophageal ulcers. Of these, twenty-five (53.2%) were with malignancy, the majority being lung cancer (seven patients) or cancer of the gastrointestinal tract (eleven patients). As compared to immunocompetent patients, immunocompromised hosts tend to have a more prolonged duration of symptoms, require a longer treatment course, and are more likely to have concomitant esophageal candidiasis [31].

#### 5.1.2. Pneumonitis

HSV pneumonitis is uncommon except in severely immunosuppressed patients, including those with hematologic malignancies and solid tumors [32,33] or in critically ill patients, especially those on mechanical ventilation [34,35]. In a cardinal paper that described 20 patients with HSV pneumonitis confirmed by autopsy findings, 17 suffered from hematologic malignancies (of whom 15 had a history of HCT) and one had a solid tumor [32]. Most common initial symptoms are dypsnea and cough as well as signs of fever and hypoxemia. Chest imaging may show focal, multifocal, or diffuse abnormalities. HSV pneumonitis can lead to respiratory failure and death; the mortality rate is estimated at >20%. Prompt treatment with antiviral therapy is key to improve outcome [33].

It should be noted that demonstration of HSV-1 (either by culture or by polymerase chain reaction, PCR) in respiratory tract specimens does not necessarily represent HSV infection of the respiratory tract, as HSV reactivation in the oropharynx leading to contamination of the sputum or bronchoalveolar lavage (BAL) samples is relatively common. Instead, confirmation of HSV pneumonitis requires histopathological or cytological examination of the clinical specimen, such as demonstration of typical HSV cytopathic effects in BAL fluid and further confirmation of the infection with immunocytochemical staining. 

#### 5.1.3. Hepatitis

HSV hepatitis is a rare condition mostly reported among third trimester pregnant women and immunocompromised patients [36]. In a review of 51 cases from the literature [36], the major underlying conditions include cancer (seven patients), renal transplant (nine patients), pregnancy (nine patients), and steroid use (ten patients). Signs and symptoms include fever, nausea, vomiting, abdominal pain, leukopenia, thrombocytopenia, coagulopathy, and marked transaminitis [36]. Due to its rarity and non-specific presentations, diagnosis is frequently missed, leading to very high mortality up to 90% when untreated. Early recognition and antiviral treatment are crucial to achieve a favorable outcome.

#### 5.1.4. Encephalitis

HSV-1 encephalitis is the most common cause of sporadic fatal encephalitis in the US and worldwide and is associated with significant morbidity and mortality despite antiviral therapy. Although HSV encephalitis is predominantly diagnosed in immunocompetent hosts, this entity may be under-recognized in immunocompromised patients. Its diagnosis in the latter group, such as those with cancer undergoing treatment or HCT, is often delayed due to atypical presentations; in particular, their immune-suppressed status may render the cerebrospinal fluid (CSF) results misleading. For example, in a case series of HSV encephalitis among seven cancer patients, CSF white blood cell (WBC) count was normal in four [16]. Others have reported similar findings of normal CSF WBC count among cancer patients when HSV-1 encephalitis was confirmed by PCR testing of the CSF [37]. While chemotherapy or corticosteroids may be the contributing factors for HSV reactivation and disease progression in cancer patients, those receiving whole-brain radiation therapy (WBRT) might be particularly at risk. HSV encephalitis has been reported in cancer patients shortly after WBRT [16]. In a recent review of 28 cases of HSV encephalitis in patients receiving chemotherapy and WBRT, seven had onset during radiation therapy, additional nine had onset within two weeks and another had onset within one month after radiation therapy [37]. Patients receiving certain immunomodulatory therapies or biologics may also be at increased risk for HSV encephalitis [38].

Patients with HSV-1 encephalitis may present with seizures, altered mental status, and fever. For those presenting with new neurological symptoms following WBRT, the possibility of HSV-1 encephalitis must be considered to prompt early initiation of antiviral therapy.

### 5.2. VZV

Immunocompromised individuals have a higher incidence of both chickenpox and HZ as well as their associated complications.

#### 5.2.1. Chickenpox or Varicella

Chickenpox is generally a benign, self-limited disease in immunocompetent children, and associated mortality is fewer than 2 per 100,000 cases, but the mortality risk increases by more than 15-fold for adults. The interval from exposure of the non-immune child to the appearance of the rash is about 14 days, with a range of 10–21 days; the incubation period may be somewhat shorter in immunocompromised children. The presenting manifestations of chickenpox are skin rash, fever, and malaise. The characteristic vesicular exanthem is often referred to as “dew drop–like” during the early stages of formation; if the vesicles do not rupture, the fluid content then becomes purulent in appearance. Skin lesions first appear on the trunk and face and rapidly spread centrifugally to involve other areas of the body. Among healthy children, crops of lesions continue to appear for about 3 days (a range of 1–7 days), with the appearance of maculopapules, vesicles, and scabs in varying stages of evolution. This phase is often prolonged in immunocompromised children, and the duration of healing the cutaneous lesions can be extended by a minimum of threefold [25]. 

Skin lesions from chickenpox can be secondarily infected by bacteria, most often with Gram-positive organisms. Other than cutaneous involvement, the most frequent complication of chickenpox is of the central nervous system, which can be manifested as acute cerebellar ataxia or encephalitis. While the former is usually benign and resolves within 2 to 4 weeks, the latter can be associated with severe outcomes. Mortality is estimated at 5–20% and neurologic sequelae is observed in ~15% of survivors [3]. Varicella associated pneumonitis is another complication that can be life threatening. This complication is more commonly observed in adults and in immunocompromised hosts, including cancer patients. In a study of 77 pediatric cancer patients with varicella, 32% had visceral dissemination and 7% died. All death was associated with primary varicella pneumonitis, with or without acute encephalitis [39].

#### 5.2.2. Shingles or Herpes Zoster

HZ complications are frequent and can present with extra disease burden in cancer patients [40]. HZ usually manifests as a vesicular rash in a dermatomal distribution. The rash is usually heralded by pain within the same dermatome by 48 to 72 h. An erythematous maculopapular rash then appears and quickly evolves into a vesicular rash, which then becomes pustular. About 20% of patients can also experience systemic symptoms such as malaise and fever. For immunocompetent hosts, the skin lesions continue to appear for 3 to 5 days, then crust by 7 to 10 days. For immunocompromised hosts, the lesions can be hemorrhagic; the rash can continue to erupt for up to 2 weeks and scabbing may not occur until 3 to 4 weeks into the disease course [3]. Immunosuppressed patients may occasionally develop a chronic cutaneous reactivation of VZV that persists for months [41]. Pain due to acute neuritis and subsequent post-herpetic neuralgia (PHN) are the most significant complications of HZ in most patients. PHN can occur in 25% to 50% of patients older than 50 years. 

While HZ can affect any dermatome, reactivation from the thoracic and lumbar ganglia are most common. Less common, but noteworthy sites of VZV reactivation include those of the cranial nerves (CN), which can be associated with significant complications. Reactivation from the first branch of the trigeminal nerve (CN V) can lead to HZ ophthalmicus; clinical findings may include conjunctivitis, keratitis, anterior uveitis or rarely, pan-ophthalmitis. When the second branch of CN V is affected, oral lesions of the palate may develop without any cutaneous lesions. VZV reactivation from the geniculate ganglion of CN VII can lead to Ramsay Hunt syndrome, which is described as a triad of ipsilateral facial paralysis, otalgia, and vesicles near the ear and auditory canal. Given the proximity of the infection to the vestibulocochlear nerve, some patients can also suffer from debilitating hearing loss, tinnitus, and vertigo. A number of neurologic complications have been described with HZ, including aseptic meningitis, encephalitis, transverse myelitis, granulomatous cerebral angiitis, and so on. When the anterior horn cells are involved, inflammation and necrosis may follow with resulting motor deficits. Stroke syndromes secondary to infection of cerebral arteries are also well-described [42] and it is important to recognize this complication when stroke symptoms develop about 6–8 weeks after the onset of HZ. While these neurologic complications may occur with or without immunosuppression, immunocompromised patients tend to have poorer outcomes. Important to note that rarely, HZ can occur without cutaneous manifestation (i.e., zoster sine herpetes) and may pose a diagnostic challenge when patients present with neuropathic pain or neurologic deficits without the typical rash [43,44].

HZ in immunocompromised patients can be more severe than immunocompetent hosts. In addition to more prolonged phase of cutaneous eruption, risk for cutaneous dissemination and visceral involvement (such as pneumonitis, hepatitis, and central nervous system disease) are higher. However, although patients with disseminated HZ may become severely ill, it is rarely fatal even in immunocompromised patients, especially with prompt initiation of effective antiviral therapy as discussed below.

## 6. Diagnostics 

### 6.1. HSV-1 and HSV-2

While a clinical diagnosis of HSV infection can be inferred when characteristic vesicular lesions are observed in oral or genital areas, herpetic lesions may also resemble ulceration from other causes. For cytologic or histopathic findings, HSV causes characteristic multinucleated giant cells and/or intranuclear inclusions, which can be demonstrated by cytologic staining methods such as Tzanck preparation. However, such a method lacks sensitivity, and cannot differentiate cytopathic changes with HSV from other herpesviruses such as VZV or CMV. Laboratory confirmation can be achieved by demonstration of viral antigens using direct florescent antibody (DFA) or viral DNA by PCR techniques.

Prior exposure to HSV can also be demonstrated by serologic assays. Most commercially available assays measure antibodies to viral glycoproteins gG1 and gG2, which are antigenically distinct between the two subtypes of HSV, allowing differential detection of HSV-1 versus HSV-2 antibodies. Of note, IgM is not very specific for primary infection. IgM was detected in 100% of the patients with primary infections of HSV-1 and HSV-2, but also in 68% of those with nonprimary infections [45]. Another study found that IgM was detected in ~30% of sera from subjects with established HSV-2 infection [46]. Thus, detection of HSV-2 IgM does not necessarily represent primary genital herpes.

### 6.2. VZV

Chickenpox and localized cutaneous shingles are commonly diagnosed by history and a physical exam that demonstrates the presence of a diffuse vesicular rash at different stages and of a vesicular rash with dermatomal distribution, respectively. However, VZV reactivation/infection in immunocompromised hosts might have an atypical presentation and are more difficult to diagnose clinically. Detection of VZV by viral culture is not clinically useful, due to low sensitivity and long turnaround time. DFA testing of scrapings from skin lesions can provide a quick diagnosis of chickenpox or shingles, but sensitivity is limited by the quality of the specimen. In the modern era, PCR has become the test of choice to confirm VZV infection in different organ systems due to its quick turnaround time and high sensitivity. PCR assays can be used for a number of clinical specimens including skin lesions, CSF, BAL, plasma, ocular fluid, lung tissues, and so on.

Serologic testing for VZV is used to establish immune status through IgG positivity; however, IgM is not useful to diagnose acute infections, as it is commonly positive in asymptomatic recurrent infections and is not specific to primary disease [5].

## 7. Antiviral Drugs and Treatment for HSV/VZV

All currently licensed systemic antiviral drugs for treatment of HSV and VZV work by inhibiting viral replication by means of inhibition of the DNA polymerase or the viral helicase-primase complex. A comprehensive review of antivirals with activities against the alpha herpesviruses is beyond the scope of this review and readers are referred to recent reviews on the subject [47,48]. Antiviral agents for systemic treatment against HSV and VZV and their major side effects are listed in Table 1 and Table 2, respectively.

### 7.1. DNA Polymerase Inhibitors

#### 7.1.1. Acyclovir and Valacyclovir

Acyclovir is a guanosine analogue and valacyclovir is an oral L-valyl ester prodrug of acyclovir with 3–5x better bioavailability as compared to oral acyclovir [49]. These drugs provide the mainstay of treatment for HSV and VZV. Acyclovir requires activation by the viral thymidine kinase into its monophosphate derivative, which is then converted into acyclovir triphosphate by cellular enzymes; the triphosphate form then functions as a competitive inhibitor and chain terminator of viral DNA polymerase.

##### Treatment of HSV Infections

Early studies have demonstrated the efficacy of acyclovir in the treatment of HSV infections. In patients with underlying leukemia, lymphoma, and HCT recipients, intravenous (IV) acyclovir 250 mg/m^2^ given every 8 h was significantly superior to placebo in treating culture-proven mucocutaneous HSV infection. On average, acyclovir treated group shed HSV virus 14 days less, had lesions scabbing and pain disappearing 4 days faster, and total healing occurred 7 days quicker compared to placebo group [50]. Similarly, adolescent and adult HCT recipients with culture-proven mucocutaneous HSV lesions treated with oral acyclovir 400 mg five times daily for 10 days had significantly less viral shedding (median 2 vs. 9 days) and formation of new lesions (1 vs. 10 days), as well as faster resolution of pain (6 vs. 16 days) and total healing of lesions (8 vs. 21 days) compared to placebo [51]. Based on these and other early studies, mild to moderate HSV mucocutaneous infection in cancer patients with intact gastrointestinal absorption are treated with oral acyclovir 400 mg three to five times daily for at least 7–14 days and severe infections are treated with IV acyclovir 5 mg/kg/dose every 8 h for at least 7–14 days and until resolution of oral lesions [2,4]. Recurrent mucocutaneous HSV lesions secondary to prolonged or recurrent chemotherapy courses are treated similarly with potential prolongation of acyclovir therapy [52]. However, due to the excellent bioavailability of valacyclovir, the use of oral acyclovir is supplanted by valacyclovir, and dosing of 500–1000 mg orally twice a day is recommended for HSV mucocutaneous and genital ulcers in immunocompromised patients [2,4].

For HSV visceral infections, it has been demonstrated that patients with solid tumors and proven, probable, or possible HSV pneumonitis had lower mortality rates when treated with HSV directed antiviral therapy [33]. Sixteen percent of patients who were treated died versus 30% of non-treated patients; further, none of the 6 patients with proven HSV pneumonia treated with IV acyclovir 10 mg/kg/dose every 8 h died [33]. There are no controlled trials for treatment of encephalitis, other visceral infections or disseminated infections caused by HSV, but higher doses of IV acyclovir at 10 mg/kg/dose every 8 h for 2–3 weeks are recommended for such severe HSV infections [53].

The most common mechanism of acyclovir resistance in HSV arises from the mutation of viral thymidine kinase, while the mutation of viral DNA polymerase is much less frequent [53]. Resistance is more common in immunocompromised hosts and is thought to constitute around 4–7% of all HSV isolates in this patient population [54]. In particular, acyclovir-resistant HSV infection is more likely to occur during the period of profound deficiency in T-cell-mediated immunity (e.g., during pre-engraftment period or graft-versus-host disease (GVHD) treatment among HCT patients) and is associated with significant morbidities. Higher doses of acyclovir prophylaxis might be needed for patients with a history of HSV during pre-engraftment or GVHD treatment. Clinically, worsening or recurrence of HSV infections in cancer patients despite adequate acyclovir prophylaxis or therapy should raise suspicion for acyclovir resistance [55]. Although some have successfully treated resistant isolates with higher doses and/or continuous infusion of acyclovir [56], early switch to an alternative antiviral (such as foscarnet) should be considered. Delay of effective therapy might lead to poor outcomes. Phenotypic testing of HSV resistance to acyclovir is commercially available; however, due to the long turnaround time (~2–3 weeks), it is not particularly useful in clinical decision making. 

##### Treatment of VZV Infections

While chickenpox and HZ are frequently self-limiting in immunocompetent patients, acyclovir therapy can substantially reduce VZV-attributed morbidity in immunocompromised hosts [57]. It has been shown to prevent dissemination of varicella in immunocompromised children [58]. Although oral acyclovir was in general effective in preventing dissemination of chickenpox in immunocompromised children, progression to pneumonitis and continuing skin lesion formation were observed with oral acyclovir [59]. Thus, IV acyclovir 10 mg/kg/dose every 8 h for 7 days is currently recommended for immunocompromised children with chickenpox, preferably if started 24 h within the onset of a rash [3]. With respect to varicella pneumonia, patients who did not receive treatment were reported to have 3.6 times higher mortality compared to acyclovir-treated patients [60].

In cancer patients with localized dermatomal HZ, oral acyclovir at a higher dose (e.g., 800 mg 5 times a day) is an acceptable antiviral agent if patient tolerates oral therapy [53]. However, valacyclovir should be the preferred agent given its superior bioavailability and less frequent dosing (e.g., 1 g 3 times daily). In cases of inability to tolerate oral formulations, disseminated cutaneous, or visceral zoster infections, IV acyclovir 10 mg/kg/dose every 8 h is recommended [53]. IV acyclovir has been shown to halt progression of localized or disseminated cutaneous HZ in immunocompromised patients, including those with hematologic malignancies, other cancers, or HCT [61]. It should be noted that acyclovir therapy aims to inhibit viral replication but is not effective in reducing the incidence or severity of PHN.

Acyclovir-resistant VZV infections are more prevalent in immunocompromised hosts and the most common mechanism of resistance is deficiency in viral tyrosine kinase [62]. However, resistance in VZV is much less common than HSV. Clinically, cancer patients with acyclovir-resistant VZV infection commonly present with unresponsive cutaneous lesions that might progress to disseminated disease. Out of 87 hematologic cancer patients with VZV infection, 59% had persistent episodes of VZV infection despite treatment with appropriate antivirals and viral resistance was identified in 27% of those with persistent episodes who were most commonly treated with acyclovir [63]. When VZV resistance is suspected or confirmed, IV foscarnet is recommended (Table 1). Cidofovir has also been used despite limited evidence. Testing for VZV resistance against acyclovir is performed in research laboratories only and no commercial testing is currently available [64].

#### 7.1.2. Penciclovir and Famciclovir 

Penciclovir is a guanosine analogue similar in mechanisms of action, activity, and spectrum to acyclovir. In a clinical trial that compared IV penciclovir to IV acyclovir in immunocompromised patients with severe mucocutaneous HSV infections requiring hospitalization, there were no differences in time regarding healing, cessation of viral shedding, resolution of pain, or treatment failure among groups that received either drug [65]. Penciclovir is licensed only for topical use (as treatment of herpes labialis) but not for systemic use.

Famciclovir is an oral diacetyl ester prodrug of 6-deoxy-penciclovir with high bioavailability. When comparing famciclovir 500 mg orally three times a day to acyclovir 800 mg orally five times a day (both given for 7 days) in immunocompromised patients with localized dermatomal HZ, there were no significant differences between the groups in the time to cessation of new lesion formation, healing of lesions, or resolution of acute phase pain. Famciclovir was well tolerated, with a safety profile comparable to that of acyclovir [66]. Famciclovir 250–500 mg three times a day could be used in cancer patients with mucocutaneous HSV infection and limited cutaneous HZ at low risk of dissemination. 

Penciclovir and famciclovir are not recommended for treatment of acyclovir-resistant HSV isolates due to the high level of cross resistance. 

#### 7.1.3. Foscarnet

Foscarnet is a pyrophosphate analogue DNA polymerase inhibitor. It is active against human herpesviruses, including HSV and VZV. Due to its significant adverse effects (including nephrotoxicity, electrolyte disturbances, genital ulceration, and others), its use for HSV and VZV is limited to treatment of severe or disseminated infection with proven or suspected resistance to acyclovir [53]. A dosing of 40–60 mg/kg every 8 h intravenously is commonly used in such settings. Topical foscarnet has also been used for treatment of mucocutaneous acyclovir-resistant HSV infections [67]. 

#### 7.1.4. Other Nucleoside or Nucleotide Analogues

##### Brivudin

Brivudin is a thymidine analogue with potent antiviral activities against HSV and VZV. The inhibitory concentration of brivudin on VZV is lower than acyclovir and penciclovir [53] and it was shown to decrease the incidence of PHN compared to acyclovir in immunocompetent patients. In patients with hematologic malignancies, oral brivudine was as effective as IV acyclovir in treating cutaneous HZ [68]. However, the metabolite of brivudine blocks the degradation of anticancer drug 5-fluorouracil (5-FU), resulting in the accumulation of 5-FU to toxic levels. Thus, it is contraindicated in cancer patients taking 5-FU or any of its prodrugs. Brivudin is licensed for treatment of HZ in several European countries but is not licensed in the US [47].

##### Cidofovir and Brincidofovir (CMX-001)

Cidofovir is a monophosphtate nucleotide analogue DNA polymerase inhibitor that does not require activation by viral kinases. It has broad antiviral activity against double-stranded DNA viruses, including the herpesviruses. Due to its toxicity profile, this drug is reserved for treatment of HSV resistant to acyclovir and/or foscarnet, commonly dosed at 5 mg/kg/dose weekly for 2 weeks followed by every other week or 1 mg/kg/dose three times a week, both intravenously. Topical cidofovir has been used in immunocompromised patients with multidrug resistant HSV mucocutaneous infections [69]. 

Brincidofovir (CMX-001) is a lipid ester prodrug of cidofovir with significantly enhanced bioavailability as well as lower rates of nephrotoxicity and myelotoxicity as compared to cidofovir. It is currently licensed for the treatment of smallpox. It has been used in the treatment of acyclovir-resistant VZV but experience is limited [70]. 

##### Ganciclovir and Valganciclovir

Ganciclovir is a synthetic analogue of 2′-deoxy-guanosine. It is available in both oral or IV formulations. Valganciclovir is a L-valyl ester prodrug of ganciclovir with significantly improved bioavailability. Ganciclovir is the first line antiviral treatment for CMV and also has activity against HSV and VZV. However, it is not generally used for the treatment of HSV and VZV infections due to increased toxicity compared to acyclovir.

#### 7.1.5. Helicase-Primase Inhibitors

Amenamevir (ASP2151) and pritelivir (AIC316) are novel oral helicase-primase inhibitors, with a new mechanism of action distinct from those involved in inhibition of the viral DNA polymerase. Amenamevir inhibits both HSV and VZV while pritelivir is active against HSV only. None of these drugs are currently licensed in the US.

##### Amenamevir

Amenamevir was shown to have equivalent efficacy for the treatment of recurrent genital herpes ulcers [71] and cutaneous HZ [72] as compared to valacyclovir and was well tolerated. While these clinical trials were performed with immunocompetent hosts, a cancer patient from Japan with cutaneous acyclovir-resistant HZ was successfully treated with amenamevir [73]. This drug is licensed in Japan for treatment of HZ.

##### Pritelivir

Pritelivir is characterized by long half-life (50 to 80 h) and can be administered once weekly [53]. In a phase 2 clinical trial, it was shown to be more efficacious in reducing viral shedding and clinical lesions in otherwise healthy men and women with recurrent genital herpes, as compared to valacyclovir [74]. A clinic trial aiming to study the efficacy and safety of pritelivir for the treatment of acyclovir-resistant mucocutaneous HSV infections in immunocompromised patients is currently on-going (ClinicalTrials.gov identifier: NCT03073967).

### 7.2. Other Treatment Considerations of HZ

In addition to antiviral therapy, good skin and wound care to prevent superimposed secondary bacterial infections are crucial for the treatment of cutaneous VZV infections. Adequate pain control is also essential for treatment of acute neuritis and PHN [3].

For PHN, multiple analgesic approaches can be employed in the chronic management of this devastating syndrome, including but not limited to opiates, antiepileptics such as gabapentin and pregabalin, tricyclic antidepressants, and topical therapies such as lidocaine and capsaicin [5]. In addition, more complex and chronic cases of PHN might benefit from multidisciplinary approaches including referrals to pain medicine specialists and psychiatry.

Concomitant steroid use with antiviral agents for the treatment of acute HZ remains controversial. In immunocompetent patients, earlier studies showed minimal improvement of quality of life and only slight benefits when adding oral steroids to acyclovir [3,75], but no clinical trial data are available for immunocompromised patients. Due to its potential side effects, steroid use is not currently recommended in this patient population for management of HZ.

## 8. Prevention

### 8.1. Antiviral Prophylaxis

Various types of cancers have been associated with increased risk of severe HSV and VZV reactivation/disease. The national comprehensive cancer network (NCCN) guidelines for prevention and treatment of cancer related infections [76] classified cancer patients at risk for HSV and VZV infections into three major risk categories: low, intermediate, and high risk. Solid tumor patients undergoing standard chemotherapy regimens are at low risk for viral reactivation and HSV prophylaxis is recommended during active therapy including periods of neutropenia only in patients who had prior HSV episodes. On the contrary, patients with lymphoma, chronic lymphocytic leukemia (CLL) and multiple myeloma are at intermediate risk of reactivation; HSV prophylaxis with acyclovir, famciclovir or valacyclovir should be considered during active therapy with the possibility of extending longer depending on degree of immunosuppression. Patients with acute leukemia are considered at highest risk for HSV reactivation and guidelines recommend prophylaxis during active induction and consolidation therapies [76].

Similar to NCCN guidelines, the Infectious Diseases Society of America (IDSA) and the American Society of Clinical Oncology (ASCO) guidelines recommend antiviral prophylaxis with nucleoside analogues (e.g., acyclovir) for HSV-seropositive patients undergoing leukemic induction therapy or allogenic HCT [77]. There was no evidence to support extending prophylaxis for more than one year as the incidence of HSV dramatically decreases 12 months after HCT [78]. 

In VZV-seropositive patients undergoing allogenic HCT, acyclovir prophylaxis significantly reduces risk of VZV reactivation, with hazard ratio of 0.16 compared to placebo, when given up to one year after transplantation [76,79]. Low dose acyclovir prophylaxis within the first year of transplantation has also shown to be effective in preventing HSV and VZV diseases in autologous HCT recipients [76]. 

Different biologics and cancer therapies also place cancer patients at heightened risk of HSV or VZV reactivation. HSV/VZV prophylaxis is recommended with the use of purine analogues (e.g., Fludarabine), anti-CD52 monoclonal antibody (e.g., Alemtuzumab), janus kinase inhibitors (e.g., Baricitinib, ruxolitinib, tofacitinib), anti-CCR4 agents (e.g., Mogamulizumab) as well as steroids (for GVHD treatment) [38,76]. Anti-CD38 agents (e.g, Daratumumab) and proteasome inhibitors (e.g., Bortezomib, carfilzomib, ixazomib) increase HZ risk and prophylaxis is recommended. TNF-α inhibitors (e.g., Adalimumab, certolizumab, etanercept, golimumab, infliximab), anti-CD20 agents (e.g., Obinutuzumab, rituximab), anti-CD30 agents (e.g., Brentuximab vedotin), and PI3K inhibitors (e.g., Idelalisib, buparlisib) can also increase risk for HSV or VZV reactivation; however, prophylaxis is not yet recommended but should be considered especially with the concomitant use of other immunosuppressive agents. Other biologic agents such as anti-SLAMF7 antibodies (e.g., Elotuzumab) or IL-6 inhibitors (e.g., Sarilumab, tocilizumab) may possibly increase the risk of HZ and prophylaxis should also be considered [38].

### 8.2. Passive Immunity

In adults, varicella zoster immune globulin (e.g., VARIZIG) is approved for post-exposure prophylaxis to prevent varicella. It should be given to at-risk individuals as soon as possible after exposure to varicella or HZ and within 10 days [80]. In a large open-label expanded-access program, high risk individuals exposed to varicella or HZ were given VARIZIG up to 10 days within exposure [81]. This study showed that postexposure administration of VARIZIG was associated with low rates of varicella in high-risk participants and it was well-tolerated. A subsequent subgroup analysis of immunocompromised individuals that included 40 adults and 263 children noted overall postexposure incidence of varicella infection at 6% and 7%, respectively, and it was concluded that VARIZIG may reduce severity of varicella in immunocompromised children and adults after exposure to varicella or HZ [82]. 

### 8.3. Vaccines

#### 8.3.1. HSV-1 and HSV-2

Currently, there are no approved vaccines for HSV. Multiple prophylactic and therapeutic vaccine trials have failed to illicit an adequate immune response against HSV-1 and HSV-2 [83]. There are multiple vaccine candidates for HSV-2 that have shown promise in animal studies but are still in early phases of clinical trials [84,85,86,87,88,89]. In addition to the vaccines based on viral antigens, mRNA-based HSV vaccines are also being developed, including BNT163 (BioNtTech, Mainz, Germany) which is currently in a Phase 1 clinical trial (ClinicalTrials.gov Identifier: NCT05432583) as well as mRNA-1608 vaccine (Moderna Inc., Cambridge, MA, USA). 

#### 8.3.2. VZV 

##### Varicella Vaccine

An Oka strain-based live attenuated vaccine is approved for varicella vaccination, as a single antigen varicella vaccine (Varivax, Merck) or in combination with measles, mumps, the rubella vaccine (MMRV vaccine) [18]. Varivax and MMRV were licensed for use in the U.S. in 1995 and 2005, respectively. Live attenuated vaccines including Varivax (as well as Zostavax, as discussed below) are in general contraindicated in immunocompromised hosts due to its potential to cause severe disease in patients who lack sufficient T-cell-mediated immune responses [90]. However, the varicella vaccine is indicated for those without evidence of varicella immunity and with sufficient time prior to immunosuppression. Per 2013 IDSA guideline [90], a 2-dose schedule of Varivax, separated by >4 weeks for patients

Aged ≥ 13 years and by ≥3 months for patients aged 1–12 years, is recommended if there is sufficient time prior to initiating immunosuppressive therapy. It should not be administered to highly immunocompromised patients and it should be administered to eligible immunocompromised patients as the single antigen product, but not combined with MMR vaccine [90]. Per current CDC recommendations, vaccination against varicella is recommended for patients with leukemia, lymphoma, or other malignancies that are in remission and who have not received chemotherapy for ≥3 months. It should also be administered 24 months after transplantation if the HCT recipient is presumed to be immunocompetent (e.g., without immunosuppressive therapy for GVHD) [18].

##### Zoster Vaccine 

Zostavax (Merck) is a live-attentuated zoster vaccine based on the Oka strain, with only limited data for efficacy and safety in cancer patients receiving chemotherapy. However, cases of Zostavax causing disseminated VZV infection with associated mortality have been reported among immunocompromised patients [91,92,93]. As of 18 November 2020, this live-attentuated vaccine is no longer available for use in the US and has been replaced by the recombinant zoster vaccine (RZV, Shingrix).

In 2017, the U.S. Food and Drug Administration (FDA) approved an adjuvanted recombinant HZ subunit vaccine (RZV, Shingrix) in adults 50 years of age and older. This approval was based on two concurrent randomized, placebo-controlled, multicentered controlled trials studying the efficacy of this vaccine in immunocompetent adults ≥50 years old [94] and adults ≥70 years old [95] with no prior history of HZ or vaccination against HZ. The first trial, ZOE-50, recruited 15,411 participants and showed an overall vaccine efficacy against HZ of 97.2% in all studied age groups (≥50 years old) [94]. The second trial, ZOE-70, focused on older adults ≥70 years old and recruited 13,900 participants with vaccine efficacy of 89.8% in preventing HZ and 88.8% in preventing PHN [95]. Right after the FDA approval, ACIP changed the zoster vaccination guidelines to include two Shingrix doses for immunocompetent people 50 years of age or older [21].

Multiple subsequent studies have shown that Shingrix induces strong and long-lasting humoral and cellular immunity against HZ [96]. Specifically, the immunogenicity and safety of Shingrix have been demonstrated in patients with solid tumors and hematological malignancies in randomized clinical trials [97,98]. One trial recruited 232 participants ≥18 years of age with solid tumors and were randomized to receive either two doses of Shingrix or placebo 1–2 months apart; they were stratified (4:1) according to the timing of the first dose with respect to the start of a chemotherapy cycle (first vaccination 8–30 days before the start or at the start [±1 day] of a chemotherapy cycle) [97]. Antibodies (humoral immunogenicity) and CD4 T-cell specific frequencies (cell-mediated immunogenicity) against glycoprotein E (the antigen in Shingrix vaccine) were significantly higher 12 months after second dose of vaccine in pre-chemotherapy and on-chemotherapy groups who received Shingrix vaccine compared to placebo. However, cell-mediated immunogenicity which is the main player in protection against VZV was better in pre-chemotherapy group compared to on-chemotherapy group. Hence, it is advisable to start Shingrix vaccine series at least one week before starting immunosuppressive agents [97]. 

Another trial recruited 606 participants ≥18 years of age with hematological malignancies who were receiving or finished immunosuppressive cancer treatments [98]. Intervention was given at least 10 days before or after cancer therapy for participants actively on therapy or at least 10 days to 6 months after finishing full cancer therapy. Around 17% of all participants had HCT at least 50 days before vaccination. Two hundred eighty-six participants were randomized to Shingrix vaccine arm compared to 283 to placebo arm. On 12-month follow up, 52% and 66.7% of participants who received Shingrix vaccine, excluding those with CLL and non-Hodgkin B-cell lymphoma, had adequate humoral and cell-mediated vaccine responses, respectively [98]. This cell-mediated vaccine response was comparable to that of immunocompetent adults. No safety concerns were significant in Shingrix vaccine group compared to placebo group in both aforementioned trials. 

A phase 2 study by Pleyer at al. showed that bruton tyrosine kinase inhibitors (e.g., Ibrutinib and acalabrutinib) in CLL patients did not significantly affect recall immune response following vaccination with Shingrix [99]. As a response to these trials, the FDA expanded Shingrix vaccine indication on 23 July 2021 to include immunocompromised adult patients 18 years of age and older who are at higher risk for HZ [100]. This also led to the new ACIP recommendations on 20 October 2021 where two Shingrix vaccines are now recommended in immunocompromised adults 19 years of age or older [101].

Currently, there is limited data about the safest timing of Shingrix vaccination after an episode of HZ disease. According to guidelines from various countries, recommendations regarding the timeframe for vaccination following an HZ are variable, ranging from after resolution of acute phase of HZ to at least 1 year [102]. These data were generated based on subgroup analysis of prior studies. An active clinical trial (ClinicalTrials.gov Identifier: NCT04091451) is recruiting adults at least 50 years of age who had prior episode of HZ and assessing safety and immunogenicity of Shingrix vaccine in this patient population.

#### 8.3.3. VZV Vaccines in Development

Other than the above mentioned licensed vaccines against VZV, there are multiple other vaccine candidates in different stages of development with at least one in a Phase 1 study (ClinicalTrails.gov Identifier: NCT03820414).

## 9. Summary 

HSV and VZV are highly prevalent viruses that can cause a wide spectrum of infections in cancer patients. Immunocompromised individuals are at increased risk of such HSV and VZV infections and their complications due to their deficient immune response. Early detection and treatment initiation are essential to limit associated morbidity and decrease mortality. A number of antiviral agents are effective in treating these infections in cancer patients, while prevention through prophylaxis and VZV vaccination are crucial in reducing incidents and severity in at risk individuals.

While these herpesviruses are important human pathogens, our understanding of their biology has also allowed their development into therapeutic tools. Genetically modified HSV-1 is now employed as oncolytic viral therapy to treat a variety of cancers, with examples including talimogene laherparepvec (T-VEC) for melanoma (FDA-approved) [103], G207 for glioblastoma (FDA fast-track designation) [104], and many others in the pipeline [105]. Thus, these viruses will continue to have significant impacts on cancer patients, as a pathogen and/or as a form of cancer therapy.

## Figures and Tables

**Table 1 viruses-15-00439-t001:** Antiviral therapy and prophylaxis for HSV and VZV infections.

HSV	VZV
**Mucocutaneous infection/esophagitis** -Acyclovir 250 mg/m^2^ or 5 mg/kg IV every 8 h, or -Acyclovir 400 mg orally 5 times per day *, or -Valacyclovir 500–1000 mg orally every 12 h, or -Famciclovir 500 mg orally every 12 h	**Chickenpox** Acyclovir 250 mg/m^2^ or 5 mg/kg IV every 8 h
**Visceral/disseminated/CNS disease** Acyclovir 500 mg/m^2^ or 10 mg/kg IV every 8 h	**Herpes Zoster** -Acyclovir 500 mg/m^2^ or 10 mg/kg IV every 8 h, or -Acyclovir 800 mg orally 5 times per day *, or -Valacyclovir 1000 mg orally every 8 h, or -Famciclovir 500 mg orally every 8 h
**Acyclovir-resistant infection** Foscarnet 40 mg/kg IV every 8–12 h, or up to 60 mg/kg every 8 h for severe disease or visceral involvment	**Acyclovir-resistant infection** Foscarnet 60 mg/kg IV every 8 h
**Acyclovir/foscarnet-resistant infection** -Cidofovir 5 mg/kg IV once weekly × 2, then every 2 weeks, or -Cidofovir 1 mg/kg IV once every other day or three times per week	
**Prophylaxis** Acyclovir 400–800 mg orally every 12 h, or Valacyclovir 500 mg orally every 12 h	**Prophylaxis** Acyclovir 400–800 mg orally every 12 h, or Valacyclovir 500 mg orally every 12 h

HSV: herpes simplex virus; VZV: varicella zoster virus; CNS: central nervous system; IV: intravenously. Modified from Dadwal SS, Ito JI. Herpes Simplex Virus Infections. In: Thomas’ Hematopoietic Cell Transplantation. John Wiley & Sons, Ltd., Hoboken, NJ, USA; 2015. p. 1078–1086; Schiffer JT, Corey L. 135—Herpes Simplex Virus. In: Mandell, Douglas, and Bennett’s Principles and Practice of Infectious Diseases, Ninth Edition. Elsevier; 2019. p. 1828–1848; and Whitley RJ. 136—Chickenpox and Herpes Zoster (Varicella-Zoster Virus). In: Mandell, Douglas, and Bennett’s Principles and Practice of Infectious Diseases, Ninth Edition. Elsevier; 2019. p. 10. * For oral therapy, valacyclovir is preferred over acyclovir given its significantly higher bioavailability and less frequent dosing schedule.

**Table 2 viruses-15-00439-t002:** Major toxicities of licensed antiviral agents for systemic treatment against HSV and VZV.

Antiviral Agents	Toxicities
Acyclovir Valacyclovir	IV formulation: Injection site phlebitis, inflammation, or vascular eruption Neurotoxicity (1–4%) Renal dysfunction (5%), most commonly due to crystal nephropathy PO formulation: Nausea, diarrhea, rash, headache, neurotoxicity, renal insufficiency Neurotoxicity, gastrointestinal upset, azotemia, localized bullous skin lesions or acute generalized pustulosis
Famciclovir	Headache, gastrointestinal upset, fatigue, rarely causes neutropenia, transaminitis, cutaneous vasculitis, rash, hallucinations, confusion
Foscarnet	Nephrotoxicity Metabolic abnormalities including hypocalcemia, hypomagnesemia, hypokalemia, hypercalcemia, hypophosphatemia, and hyperphosphatemia Seizures, dystonia, headache, tremor, hallucinations Fever, rash, gastrointestinal upset, transaminitis Hemorrhagic cystitis, painful genital ulcers
Cidofovir Brincidofovir	Nephrotoxicity, neutropenia (24%), fever, rash, gastrointestinal upset, iritis, uveitis Diarrhea
Ganciclovir	Myelosuppression mainly neutropenia (up to 40%), headache, confusion, seizures, coma, rash, phlebitis, fever, transaminitis, gastrointestinal upset
Brivudin (licensed in few European countries only, not in the US)	Gastrointestinal upset, drug-induced hepatitis, headache, dizziness, delirium, major drug–drug interaction with 5-Fluourouracil
Amenamevir (licensed in Japan only, not in the US)	Liver toxicity and renal disorder with higher doses, headache, thrombocytopenia

IV: intravenous; PO: per os; US: United States.

## Data Availability

Not applicable.

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
