# Peer review of "Herpes Simplex Virus and Varicella Zoster Virus Infections in Cancer Patients"

_viruses, 2023, doi:10.3390/v15020439_

Round 1

Reviewer 1 Report

The review article is written about the Herpes simplex virus & Varicella Zoster virus, its pathophysiology, diagnostics, and currently available treatments. However, in my opinion, this review article has some shortcomings regarding some sections. I feel that the title of this review must be modified as there is no dedicated section for HSV and VZC either in cancer or different types of cancers. Also, the author claims that HSV and VZV infections are more prone to immune compromised patients. In addition, I have a few remarks listed below:

1.     Authors did not provide the biology of HSV and VZV viruses

2.     Vaccine section, 7.3.1. could be expanded, can list out the most promising pre-clinical vaccines for HSV1 and HSV2, m-RNA-based vaccines for HSV etc,.

3.     A graphical representation summarizing HSV/VZV infection in cancer(s) is recommended.

4.     In Section 7.3.2, remove the VZV vaccines in development, which is separated from VZV the vaccine section.

5.     In Section 6.1.1, Resistance is more common in immunocompromised hosts and is thought to constitute around 4-147% of all HSV isolates in this patient population.

Given these shortcomings, this review article requires major revisions to reconsider.

Author Response

Dear Reviewer 1,

Thank you for taking the time in reviewing our manuscript. Please find attached a point-by-point response to the comments. 

Reviewer 2 Report

This paper reviewed of HSV and varicella zoster virus in cancer patients. However, most of the full text discussed the herpes virus and varicella zoster virus, and did not focus on cancer patients. For example, in epidemiology, in addition to the infection of HSV, the infection of HSV in tumor patients should be discussed. In particular, HSV is a good oncolytic virus vector, and two oncolytic virus products with HSV-1 as the carrier have been proved (T-vec, Amgen; G207, Daiichi Sankyo). Therefore, whether there is HSV antibody positive in tumor patients, especially in patients with melanoma and glioma, and whether the HSV antibody has an impact on the treatment of tumor oncolytic virus are discussed.

In the third part, Immunology and host responses, it is incorrect to juxtapose NK cells, macrophages, DC cells and Toll like receptor pathways. Different tumors have different heterogeneity, resulting in different immune microenvironments of different tumors. Therefore, the infection of HSV or VZV in different tumor patients needs to be discussed separately, and whether infection will cause differences in gene or cell therapy methods for tumors. This is very important for the infection of HSV and VZV in tumor patients. In the fourth part, the pathogenesis and clinical manifestations, a simple description of the infection process of HSV and VZV occupies too much space. The author should more review the description of the infection process of HSV and VZV in tumor patients (taking 2-3 different types of tumors as examples), especially whether the receptor and downstream signal pathway of target cells are different from those of infected normal people. The fifth part is diagnosis. For the same problem, it is suggested to review the fluorescent quantitative PCR method and TCID50 method used by the proved oncolytic virus in the treatment of melanoma for the detection of HSV virus DNA and live virus, and review the use of HSV for virus shedding detection in melanoma patients and PCR method in normal people, so as to highlight the significance of HSV diagnosis in tumor patients. The seventh part, prevention, can introduce the reasons for the failure of existing vaccines against HSV and the poor immunogenicity of HSV itself, which can not induce high titers of neutralizing antibodies. At the same time, it is recommended to discuss the research of mRNA vaccine against HSV virus, because many clinical trials of mRNA vaccines against HSV have shown that they have good neutralizing antibody titers.

There are many problems in the literature citation of the full text. In many places, there are no references or insufficient references.

Author Response

Dear Reviewer 2,

Thank you for taking the time to review our manuscript. 

Please find attached a point-by-point response to the comments. 

Round 2

Reviewer 2 Report

The manuscript has been sufficiently improved compared with the previous version. The current version has met the requirements of publication.